# Spatiotemporal Dynamics of Ecological Condition in Qinghai-Tibet Plateau Based on Remotely Sensed Ecological Index

Jiaxi Cao [1,†], Entao Wu [1,†], Shuhong Wu [1,*], Rong Fan [1], Lei Xu [1], Ke Ning [2], Ying Li [1], Ri Lu [1], Xixi Xu [1], Jian Zhang [1], Junliu Yang [3], Le Yang [1,4] and Guangchun Lei [1]

1   School of Ecological and Nature Conservation, Beijing Forestry University, Beijing 100083, China
2   School of Statistics, Jilin University of Finance and Economics, Jilin 130117, China
3   School of Soil and Water Conservation, Beijing Forestry University, Beijing 100083, China
4   Tibet Plateau Institute of Biology, Lhasa 850000, China
*   Correspondence: wshuhong@bjfu.edu.cn; Tel.: +86-010-136-9120-0800
†   These authors contributed equally to this work.

**Abstract:** The eco-system in the Qinghai-Tibet Plateau (QTP) is extremely fragile, and highly vulnerable to climate change. Knowledge of the changes in the ecological conditions is vital to mitigate the impact of climate change. In this study, we investigated the trend of ecological conditions of the QTP using the remotely sensed ecological index (RSEI), which is the first PCA (principal component analysis) axis of the four indexes derived from the MODIS (Moderate resolution Imaging Spectroradiometer) images captured in the growing season of 2000–2020. The four indexes, i.e., NDVI (normalized difference vegetation index), heat (land surface temperature, LST), wetness (tasseled cap wetness index, WET) and dryness (normalized difference impervious surface index, NDBSI), were calculated on the Google Earth Engine platform. Using land use cover change (LUCC) data, long-term meteorological records and the supplementary annual livestock production, we explored the drivers of spatiotemporal changes in the RSEI. The results show the following points: (1) the ecological conditions of the QTP have remarkable spatiotemporal variations. There were two ecological degradation periods, one of them occurred in the central region during 2005–2010, mainly attributed to the rising temperatures and decreasing precipitation. The other occurred during 2015–2020, driven primarily by overgrazing in the southwest. From 2000 to 2005, it was a period of rapid ecological restoration in the ecologically fragile northeast region. (2) The contribution rate of pc1 was stable at 60%, while the contribution rate of pc2 remained below 40%, indicating that pc1 demonstrated most of the characteristics of the four indexes. The correlation coefficients between NDVI and WET with pc1 are both positive, while LST and NDBSI have negative correlation coefficients, i.e., negative effects. This is consistent with the actual situation. (3) Overgrazing caused grass degradation in the southwest area of the QTP, which might be the main reason for the poor ecological conditions (i.e., low RSEI value) during 2015–2020. (4) Temperature and precipitation showed an increasing trend during the study period. A warmer and wetter climate is expected to have profound impacts on the ecosystems in QTP and practices should be concentrated on identifying climate-sensitive ecosystem components and designating adaptative options.

**Keywords:** ecological condition evaluation; alpine environment; MODIS; climate change; adaptive management; RSEI



## 1. Introduction

Ecosystems provide fundamental life-support services for human well-being and social development [1]. However, in the process of modernization, rapid urbanization has directly or indirectly led to land use changes [2], which in turn result in increasing pollution and ecosystem degradation [3,4], especially in arid and semi-arid areas with

low ecological resilience [5]. Loss of ecosystem integrity is mainly manifested in land degradation, sharp reductions in natural vegetation cover, desertification, and increases in the extent and degree of salinization at the landscape scale [6,7]. These phenomena strongly restrict and affect the sustainability of social and economic development, and may even threaten the survival of mankind [8]. Therefore, sound understandings of the long-term ecosystem integrity and its drivers are prerequisites for safeguarding ecosystem functions and services, and for long-term human well-being [6,9]. This is particularly critical for ecosystems in regions with fragile environments, as they have low resilience to anthropogenic disturbances and low capacities to recover [10].

Many approaches have been proposed and used to quantify and assess ecosystem integrity [11], or more generally, regional ecological conditions [12,13] with measurements and indices of physical, chemical, biological properties and community functional and structural attributes [14]. Generally, ecological condition assessment is based either on in-situ measurements or surveys [15], semi-quantitative expert opinions [16], and/or remote sensing techniques [13,17]. While ground surveys and measurements are often applied to specific areas of aquatic or terrestrial ecosystems (e.g., [18,19]), remote sensing techniques are applicable to the entire region, comprising both aquatic and terrestrial ecosystems [13]. In practice, multiple approaches are combined in an assessment [14] to produce an unbiased and robust result.

Satellite images, such as Sentinel-2, Landsat, and MODIS, are suitable for mapping land cover land use patterns [20], for measuring vegetation conditions and structures [21], as well as for estimating landscape fragmentation and degree of anthropogenic pressure [22]; therefore, they are widely used to assess ecosystem integrity and environmental quality. The remote sensing ecological index (RSEI, proposed in [23]), an environmental quality assessment tool that integrates several satellite-derived indices, can be applied at a variety of spatial and temporal scales; thus, it can be used to explore the spatiotemporal dynamics of ecological conditions. Land use cover change (LUCC) is one of the most important drivers of natural environment quality change and key components of ecosystem integrity [2]. While causing significant changes in the surface structure, LUCC also directly affects the regional atmosphere, hydrology, soil, biodiversity and biogeochemical cycle processes, thereby changing the overall structure and composition of the ecosystems in a region [24]. The analysis of LUCC can further interpret the spatiotemporal dynamics of RSEI.

The Qinghai-Tibet Plateau (QTP) contains some of the most fragile ecosystems in the world [25]. There is sufficient cause to believe that alpine grasslands in QTP may be particularly sensitive to anthropogenic disturbances, especially to the scale of over-grazing that has been taking place over the last few decades [26]. Moreover, many ecosystems in the region, such as alpine wetland and mountainous forests, are extremely vulnerable to climate change [27,28]. Many studies have demonstrated that the average temperature has risen and precipitation has increased in QTP over the past few decades (see [29]). These warmer and wetter trends could exert greater pressure on the local ecosystems and lead to further losses of ecosystem integrity [29]. In the meantime, the unique alpine ecosystem is a vital section of China's natural resources and plays an important role in maintaining regional and national biodiversity, as well as ecological integrity [30,31]. Slight changes in the QTP may lead to the fluctuation of thermal and dynamic processes in this area [32], and this change could lead to ecological catastrophes in the surrounding area [33].

Although past research on climate change in QTP has accumulated extensive knowledge on climate trends, the study of ecosystems' conditions and adaptive mechanisms are starting to emerge, and the knowledge on spatiotemporal dynamics is particularly limited.

During the construction of ecological civilization, the QTP has attracted much attention because of its important ecological status. However, its sparsely populated condition makes it difficult to conduct a comprehensive survey with fieldwork. The insufficient survey results are not enough to support the evaluation of the overall ecological conditions in the QTP. Therefore, using data from satellites to establish evaluation models is the

preferred method. RSEI has been widely used in the ecological condition assessment of complex ecosystems since its inception. However, the original RSEI construction requires downloading satellite data, processing and superimposing operations one by one. This process creates some systematic errors and takes a significant amount of time. This study integrates data acquisition, data preprocessing, error analysis, and result plotting on the GEE (Google Earth Engine) platform. After the QTP is identified as the study area, the calculation results of RSEI can be obtained directly from the GEE, which will improve the accuracy of the evaluation, while saving research time.

In this study, four MODIS (Moderate Resolution Imaging Spectroradiometer) indices, namely NDVI (normalized difference vegetation index), wetness (tasseled cap wetness index, WET), LST (land surface temperature), and NDBSI (normalized difference impervious surface index) in 2000, 2005, 2010, 2015 and 2020 of the QTP were calculated on GEE. The four indices were integrated into a single RESI based on a principal component analysis (PCA), and the changes in RSEI in the four periods of 2000–2005, 2005–2010, 2010–2015 and 2015–2020 were investigated and mapped. Finally, the contributions of LUCC, agricultural production, and climate (temperature and precipitation) to the mapped spatiotemporal patterns were identified.

## 2. Materials and Methods

In this study, RSEI [23] was used as an evaluation index to explore the environmental conditions in the QTP during 2000–2020. RSEI was calculated at 5-year intervals. After obtaining the spatiotemporal distribution characteristics of RSEI, the contribution rate of the constituent indexes was analyzed using PCA. Trend analysis is used to assess the trends in the environmental conditions in the QTP. The reasons and drivers of the spatial-temporal distribution patterns of RESI are analyzed with LUCC, temperature, precipitation, grazing and human activities.

### 2.1. Study Site

The Qinghai-Tibet Plateau (73°19′ E–104°47′ E, 26°00′ N–39°47′ N) stretches from the southern edge of the Himalayas in the south to the northern edge of the Qilian Mountains in the north, to the Pamir Plateau and the Karakoram Mountains to the west, and to the northeast by the Chin Ling Mountains and the Loess Plateau (Figure 1).

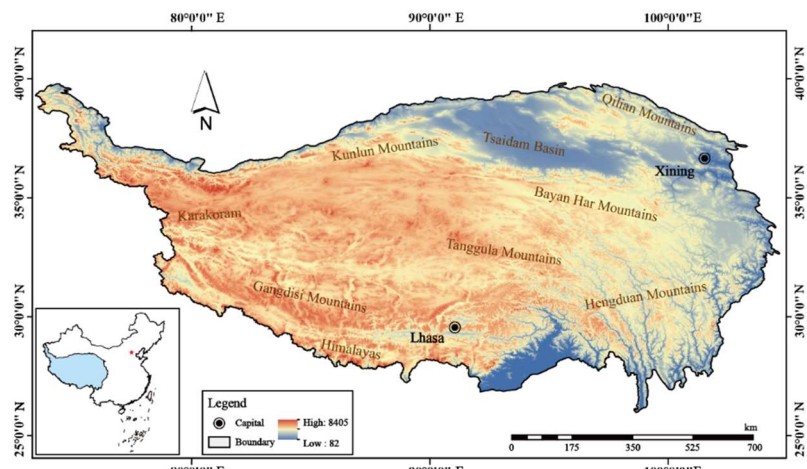

**Figure 1.** The Qinghai-Tibet Plateau covers about $2.5 \times 10^6$ km$^2$. The mean elevation is about 4500 m above sea level and is the head water of many large rivers in China.

Because of its high altitude and undulating terrain, it has become the source of many rivers. The southern and eastern marginal mountainous rivers are densely packed, with larger outflows that belong to the Indian Ocean system of the Brahmaputra and Nu, and the upper reaches of the Yangtze, Yellow and Lancang rivers that belong to the Pacific



Ocean system [34]. The area is generally between 3000 and 5000 m above sea level, with an average altitude of more than 4000 m. The distribution of rain and heat is extremely uneven, with the average temperature in the southeast region being 20 °C and decreasing to below −6 °C in the northwest. The annual precipitation in the south is 1000–4000 mm, but in the west, it is only 20–100 mm [35].

The logical flow of this study is shown as (Figure 2). The temperature and precipitation data are downloaded from the China Meteorological Data Network. The data of livestock, animal husbandry products and ecological governance status are obtained from local statistical yearbooks and policy documents.

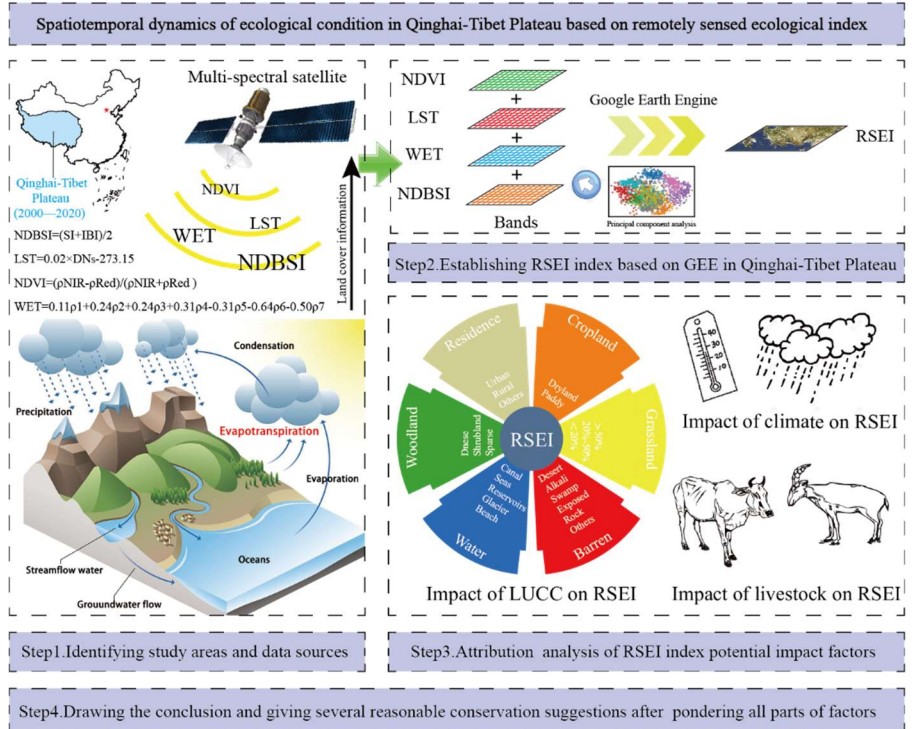

**Figure 2.** Flowchart of the study. Firstly, the Qinghai-Tibet Plateau is selected as the research area, and the data needed to construct RSEI are all from MODIS. At the same time, the land cover information and biogeochemical cycle were obtained. After that, RSEI was obtained by PCA analysis using NDVI, LST, WET, NDBSI on GEE. To analyze the drivers of the spatial and temporal distribution characteristics of RSEI, LUCC, temperature, precipitation, animal husbandry and human factors were discussed.

### 2.2. Data and Source

RSEI (remote sensing ecological index) is the combination of four indices, namely NDVI (normalized difference vegetation index), wetness (tasseled cap wetness index, WET), heat (land surface temperature, LST) and dryness (normalized difference impervious surface index, NDBSI). MODIS (Moderate Resolution Imaging Spectroradiometer) products, including MOD13A1, MOD11A2, MOD09A1 and MOD09A1, are used to calculate the four indices on GEE. All MODIS satellite images are from the growing season (July–September) and its spatiotemporal resolution is shown in Table 1.

The data of land use cover change (LUCC) are obtained from the data sharing platform of the Resource and Environment Science and Data Center in the Chinese Academy of Sciences. LUCC is based on the US Landsat remote sensing image products. This dataset is obtained with image fusion, geometric correction, image enhancement, machine supervised classification, manual visual interpretation and field verification, which is mainly for terrestrial ecosystems. This division system includes 6 class I elements (such as cropland, woodland, residence, barren, grass and water) and 25 class II elements.

**Table 1.** Indices for computing RSEI.

| Index | Product | Spatial Resolution | Temporal Resolution | Level |
|---|---|---|---|---|
| NDVI | MOD13A1 | 1000 m | 16 d | L3 |
| LST | MOD11A2 | 1000 m | 8 d | L3 |
| WET | MOD09A1 | 500 m | 8 d | L3 |
| NDBSI | MOD09A1 | 500 m | 8 d | L3 |

*2.3. RSEI*

RSEI is mainly calculated in two steps [23]. NDVI, WET, LST and NDBSI, representing the easily interpreted greenness, wetness, heat and dryness aspects, respectively, are first calculated. The loading of each index to RSEI at the same time is determined by principal component analysis (PCA). Ultimately, RSEI is calculated as Equation (1).

$$RSEI = f(NDVI, WET, LST, NDBSI) \tag{1}$$

The larger the RSEI index, the better the quality of the environmental condition. The method of RESI aims to improve the availability of ecological evaluation indicators, to eliminate the interference of human subjective consciousness on factor assignment, and widely use medium-resolution satellite data, which effectively expands the scope.

Recently, NDVI has been widely used in the description of vegetation properties, such as regional vegetation coverage, plant biomass and leaf area, and is also an important ecological indicator [36]. It is calculated as Equation (2).

$$NDVI = \frac{\rho_{NIR} - \rho_{Red}}{\rho_{NIR} + \rho_{Red}} \tag{2}$$

where $\rho_{Red}$ and $\rho_{NIR}$ is spectral reflectance in the red and near-infrared bands, respectively.

Temperature is an important factor that affect vegetation growth, and it is also one of the important indicators driving environmental change [37]. In this study, the gray value of the remote sensing data is converted into the Celsius scale, and then the distribution of surface temperature in the study area is obtained and LST can be calculated as Equation (3).

$$LST = 0.02 \times DN_S - 273.15 \tag{3}$$

where $DN_S$ is the grayscale value of the land surface temperature image.

Studies [38] have shown that the humidity component of the tassel cap transformation in MODIS can well reflect the comprehensive humidity of soil and vegetation in the QTP. Therefore, in this study, the tassel cap transformation of Equation (4) is combined with the surface reflectance product of MOD09A1 to calculate the humidity index (WET), which is as follows:

$$WET = 0.1147\rho_1 + 0.2489\rho_2 + 0.2408\rho_3 + 0.3132\rho_4 - 0.3122\rho_5 - 0.6416\rho_6 - 0.5087\rho_7 \tag{4}$$

where $\rho_i (i = 1, 2, \ldots, 7)$ is the surface reflectance product reflectance of each band.

As the process of urbanization gradually accelerates, the area of impervious layers is increasing year by year, while the environmental damage caused by economic activities is gradually expanding the area of bare natural soil, which together lead to surface drying [39]. In this study, the bare soil index (SI) and the normalized difference built-up index (IBI) were used in the weighted average to obtain the NDBSI as Equations (5)–(7).

$$SI = \frac{(\rho_{SWIR1} + \rho_{Red}) - (\rho_{Blue} + \rho_{NIR})}{(\rho_{SWIR1} + \rho_{Red}) + (\rho_{Blue} + \rho_{NIR})} \tag{5}$$

$$IBI = \frac{\frac{2\rho_{SWIR1}}{\rho_{SWIR1}+\rho_{NIR}} - \left( \frac{\rho_{NIR}}{\rho_{NIR}+\rho_{Red}} + \frac{\rho_{Green}}{\rho_{Green}+\rho_{SWIR1}} \right)}{\frac{2\rho_{SWIR1}}{\rho_{SWIR1}+\rho_{NIR}} + \left( \frac{\rho_{NIR}}{\rho_{NIR}+\rho_{Red}} + \frac{\rho_{Green}}{\rho_{Green}+\rho_{SWIR1}} \right)} \qquad (6)$$

$$NDBSI = \frac{SI + IBI}{2} \qquad (7)$$

where $\rho_{SWIR1}$, $\rho_{Red}$, $\rho_{Blue}$, $\rho_{Green}$ and $\rho_{NIR}$ are the spectral reflectance in the shortwave infrared, red, blue, green, and near-infrared bands, respectively.

According to the value from high to low, RSEI can be divided into five levels with equal distance in ArcGIS, namely best, good, normal, bad and worst, respectively [40].

### 2.4. Data Processing

In this step, all four indices, namely NDVI, LST, WET and NDBSI, in 2000 are taken as examples. The data sources are MOD13A1, MOD11A2, MOD09A1 and MOD09A1, and the acquisition interval of images are 16 days, 8 days, 8 days and 8 days, respectively. All the MODIS satellite images are from the growing season (July–September) in QTP. After noise reduction, geometric correction, image enhancement and grayscale processing, a total of 4, 7, 7 and 7 images were obtained in 2000 for NDVI, LST, WET and NDBSI, respectively. All indices are averaged as the amount for this duration. After obtaining the four means, the RSEI of the current year is calculated after PCA analysis. Each year is calculated in the same way; a total of five RSEIs were performed in 2000, 2005, 2010, 2015, and 2020, respectively. After the cloud removal of each image in the research period, the median value of each image was taken. The cloud coverage was 2.37% in 2000, 3.47% in 2005, 2.00% in 2010, 0.17% in 2015, and 0.30% in 2020, respectively. All steps are implemented in GEE, and the operator needs to enter the vector range of the study area. The data acquisition and processing are carried out automatically by the GEE. After the program stops, RSEI spatial characteristics and assignments for the study area and the proportion of each index in the calculation of the current year will be obtained.

### 2.5. Trend Analysis

Trend analysis is defined as a process of estimating the gradual change in future events from past data. Different parametric and nonparametric techniques are used to estimate trends [41]. Subsequently, it has evolved as a statistical technique to estimate the gradual change in the time series, and predict the future data changes [42]. After the result of RSEI is worked out by the GEE platform, a raster calculator tool in ArcGIS is used to obtain the change trend of RSEI for the four periods with the differences in the raster [43]. The results were divided into better (>0), unchanged (=0), and worse (<0), which were characterized as ecological recovery, stability, and degradation, respectively [43]. Thus, the trend of ecological change in the QTP during 2000–2020 was comprehensively analyzed.

## 3. Results

### 3.1. PCA

The results of PCA (Table 2) show that in the five calculations during 2000–2020, the contribution rate of pc1 was stable at 60%, while the contribution rate of pc2 remained below 40%, indicating that pc1 demonstrated most of the characteristics of the four indexes. All four indexes contribute to pc1 and are relatively stable. The correlation coefficients between NDVI and WET with pc1 are positive, indicating that they have a positive effect on the calculated value of RSEI. LST and NDBSI have negative correlation coefficients, i.e., negative effects, which are consistent with the actual situation. In addition, NDVI and pc1 demonstrate a strong positive correlation relationship, with correlation coefficients between 0.4 and 0.8. LST and pc1 have long been strongly negatively correlated, with correlation coefficients between −0.9 and −0.6. To sum up, pc1 can maximize the concentration of the characteristics of each index and reasonably explain ecological phenomena, so it can be used to create RSEI. Combined with the results of LUCC and climate data analysis, it is clear that there is an increase in the contribution rate of WET and a decline in LST in 2020,

which are related to the increase in temperature and glacial meltwater in the QTP from 2015 to 2020. This change will lead to an increase in water content and a decreasing trend in surface heat. The reason behind this phenomenon will be discussed in the trend analysis.

**Table 2.** Loadings of each index to the first axis of PCA for the studied years.

| Index | Years | | | | |
|---|---|---|---|---|---|
| | **2000** | **2005** | **2010** | **2015** | **2020** |
| NDVI | 0.4809 | 0.6411 | 0.7508 | 0.6945 | 0.6742 |
| LST | −0.8702 | −0.7559 | −0.6436 | −0.703 | −0.6662 |
| WET | 0.106 | 0.1323 | 0.1486 | 0.1534 | 0.3189 |
| NDBSI | −0.0004 | −0.0001 | −0.0001 | −0.0009 | −0.001 |
| EV (pc1) | 0.0481 | 0.0572 | 0.0562 | 0.0572 | 0.0671 |
| EV (pc2) | 0.0359 | 0.0319 | 0.032 | 0.0239 | 0.0249 |
| ECR (pc1%) | 56.1 | 62.54 | 61.75 | 68.25 | 66.73 |
| ECR (pc2%) | 41.92 | 34.87 | 35.12 | 28.51 | 24.77 |

Note: Default NDVI, LST, WET, NDBSI display correlation with PC1; EV (pc1), EV (pc2), ECR (pc1%) and ECR (pc2%) are the abbreviations of eigenvalue, principal component 1, principal component 2, contribution rate of the first principal component and contribution rate of the second principal component, respectively.

*3.2. RSEI*

Generally, the spatiotemporal characteristics of RSEI show that the ecological conditions of QTP are predominantly normal and good. The year 2015 had the lowest proportion of land with normal and good characteristics (i.e., 67.51%, Figure 3n), and the proportion reached the maximum of 79.29% in 2000 (Figure 3k). The proportion of land with normal and good characteristics stabilized at about 70% for the rest of the tested years, indicating that the ecological conditions of the QTP are generally good. From the perspective of the spatial distribution characteristics of RSEI, the QTP can be divided into three parts, the northern, the southwest and the southeast.

The northern region has the largest area with the worst level for 20 years. This region belongs to the Tsaidam Basin, where the surface temperature is high and both surface and groundwater resources are scarce. The vegetation in the region is sparse, and a large proportion of the land is barren. RSEI shows that this area is ecologically fragile and needs appropriate conservation measures to prevent ecological degradation. From the perspective of temporal characteristics, the proportion of bad land decreased from 5.39% (Figure 3k) in 2000 to 4.91% (Figure 3l) in 2005, and has remained below 5% in the subsequent period, suggesting that the past management in this area is effective.

The normal and bad land are mainly located in the southwest region, and the proportion of bad land decreased from 15.1% (Figure 3k) to 13.92% (Figure 3l) in 2000–2005, indicating that the ecological conditions of the region improved during this period. However, the proportion of bad land increased to 20.11% (Figure 3m) in 2010 and remained steady over the subsequent periods, indicating that the ecological conditions in this area have been degrading since 2010.

The best land and good land are mainly distributed in the southeastern region. In 2010, the proportion of the best land was at its minimum at 0.02% (Figure 3m) and reached a maximum of 6.67% (Figure 3n) in 2015. The proportion of good land was the lowest (27.58%, Figure 3m) in 2010 and highest (43%, Figure 3l) in 2005.

The prevailing ecological conditions of this area are good, probably due to the lower altitude. There are few human activities and the hydrothermal conditions are good, which are suitable for vegetation growth. Vegetation growth, in turn, conserves water to feed back into the ecosystem [44].

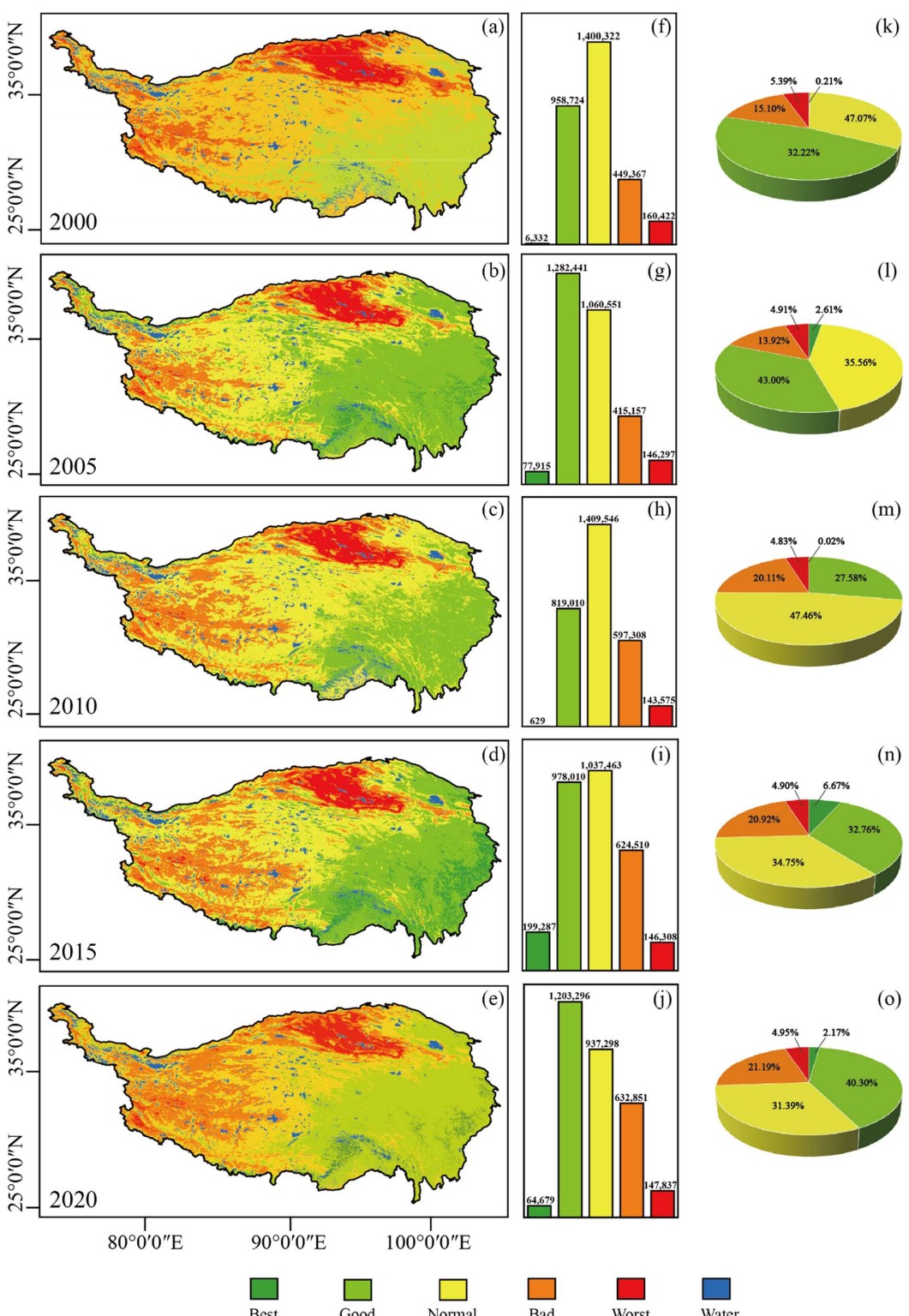

**Figure 3.** Spatiotemporal patterns of RSEI in the Qinghai-Tibet Plateau. RSEI is divided into 5 levels: best, good, normal, bad and worst according to the value with equal distance in ArcGIS [40]. (**a**–**e**) show the temporal and spatial distribution characteristics of RSEI in QTP. (**f**–**j**) show the total area (km²) of best, good, normal, bad and worst levels in each studying period. (**k**–**o**) show the proportion of best, good, normal, bad and worst levels in each studying period.

### 3.3. Trend Analysis

The results show that there is a large spatiotemporal difference (Figure 3a–e) in the RSEI, and only the analysis of its multi-period status cannot obtain the change in ecological conditions. Therefore, the raster calculator of ArcGIS is introduced to treat the RESI results of adjacent time as a difference. The visualization of the spatiotemporal changes in RESI is shown in (Figure 4)

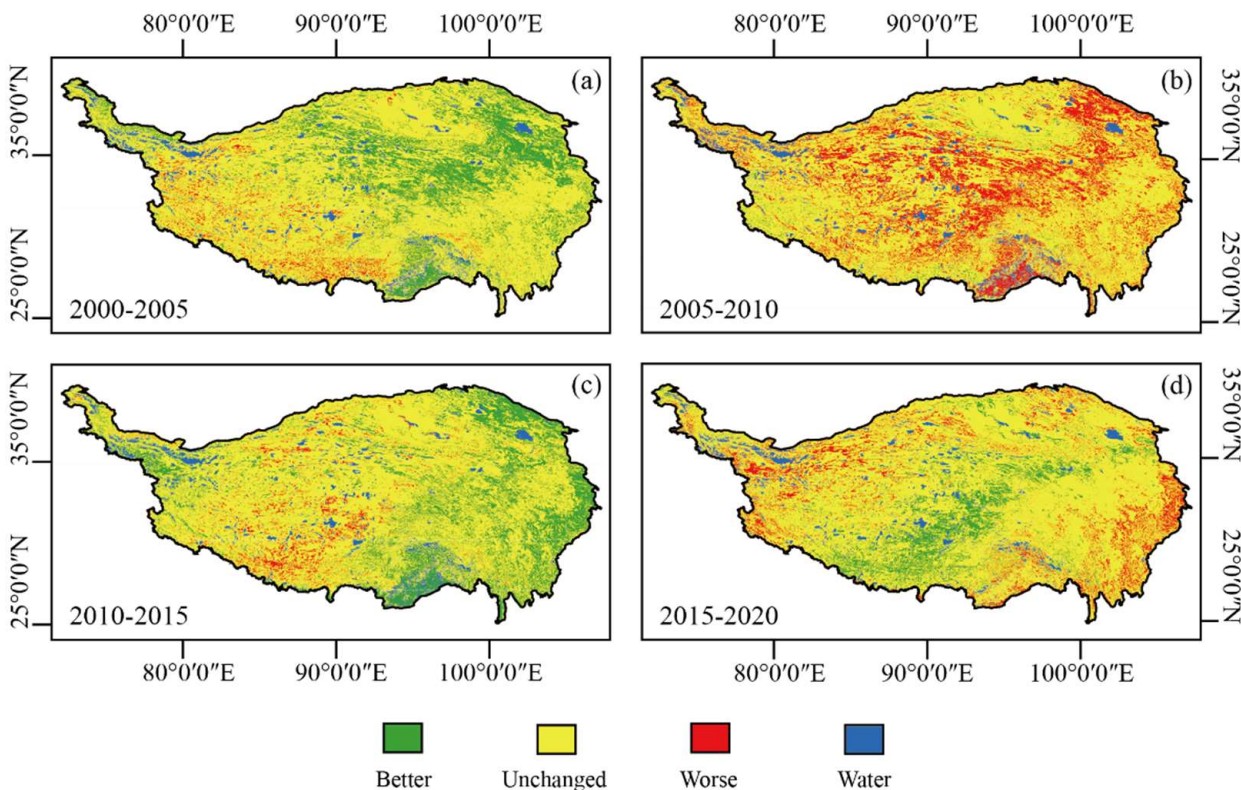

**Figure 4.** Spatiotemporal patterns of trend analysis of RSEI. (**a**–**d**) show the trend of temporal and spatial distribution characteristics of RSEI in QTP. Better, unchanged and worse indices respectively indicate that the RSEI of the current year on QTP is better, unchanged and worse than that of the previous period.

The calculation results (Table 3) show that the sum of the better and unchanged indices in the QTP during 2000–2005 was 95.69%, indicating that the period was in ecological recovery. This value fell to 72.45% during 2000–2005, of which the proportion of the better index fell to 1.21%, indicating that it was in ecological degradation during this period.

**Table 3.** Result of trend analysis of RSEI in the Qinghai-Tibet Plateau.

| Index | 2000–2005 | 2005–2010 | 2010–2015 | 2015–2020 |
|-------|-----------|-----------|-----------|-----------|
| Better | 20.85% | 1.21% | 21.93% | 10.70% |
| Unchanged | 74.85% | 71.24% | 73.18% | 77.57% |
| Worse | 4.31% | 27.56% | 4.89% | 11.74% |

During 2010–2015, the sum of the better and unchanged indices rose to 95.11%, of which the proportion of the better index rose to 21.93%, which was the maximum, indicating that the quality of the ecological environment was in a state of recovery during this period. During 2015–2020, the value remained at 88.27%, of which the proportion of the better index dropped to 10.70% compared with the previous stage, indicating that the ecological conditions in the QTP was in a slow state of recovery. It should be noted that the proportion

of the worse index reached the maximum of 27.56% during 2005–2010, indicating that the quality of the ecological environment declined sharply during this period.

From a spatial perspective, most of the region remained stable during 2000–2005, with slight degradation in the southwest and recovery in the central and southern areas (Figure 4a). During 2005–2010, the north-east, central and southern regions experienced a period of degradation, and the analysis concluded that rising temperatures and reduced precipitation were the main drivers of this trend (Figure 4b). During 2010–2015, the ecological conditions of this area were stable, with sporadic declines in the southwest and western regions (Figure 4c), and the marginal areas recovered from the previous period. During 2015–2020, the central and southern regions were in a state of recovery, and the western and eastern areas were in a state of sporadic degradation.

In summary, during the entire study period from 2000 to 2020, the proportion of the unchanged index remained at 71.24–77.57%, indicating that the ecological conditions of the QTP remained stable. Combined with the dynamic changes in the proportion of the better and worse index, it can be found that the region shows a change pattern of recovery–degradation–recovery–slightly degradation. Ecological condition degradation is concentrated in the 2005–2010 and 2015–2020 periods, and the drivers will be discussed in the subsequent sections.

## 4. Discussion

### 4.1. Drivers of RSEI Dynamics in QTP

#### 4.1.1. LUCC

The dominated land cover in the QTP include the barren and grass areas [45]. The grass areas are mainly distributed in the southwest and central regions of the QTP, and their proportion has remained above 50% from 2000 to 2015, with a maximum of 58.50% (Figure 5k) in 2000 and a minimum of 48.87% (Figure 5o) in 2020. A combination of human and natural factors contributed to the degradation of grass areas during this period [46]. The barren areas were mainly distributed in the northern region and to a lesser degree in the southern region during 2000–2015. Their proportion is stable at around 25%, peaking at 32.55%. Compared with the period of 2000–2015, a large area of barren land appeared in the southwest of the QTP in 2020 (Figure 5e), which was degraded in the same period as for the grass areas.

Woodland areas, which are the third largest land cover, are mainly distributed in the southern low-altitude areas. Their proportion stabilized at about 10% from 2000 to 2015, rising to 12.37% in 2020 (Figure 5o). Woodland areas can be divided into dense, shrubland and sparse categories from the perspective of forest stand composition. Combined with the imaging principle of satellite imagery, there might be a conversion from dense to sparse. This conversion may suggest a degradation in QTP during this period.

The fourth rank is water, which is widely distributed throughout the QTP with minimal distribution characteristics [32]. Its proportion stabilized at around 4.2% from 2000 to 2015 and rose to 5.26% in 2020 (Figure 5o). This trend that might be related to the warm humidification of the QTP and the increase in glacial meltwater during this period. Cropland is the fifth rank, which is stable in the range of 0.7–0.9% during the entire study period with minimal distribution characteristics. From 2015 to 2020, the cropland areas increased by about $2 \times 10^4$ km$^2$ (Figure 5i,j), which has a minimal impact on the ecological conditions. The lowest proportion of all the I classes was residence, which accounted for less than 0.1% for many years.

The components of LUCC in the QTP did not change much during the period from 2000 to 2015. The grass areas decreased by about $2.4 \times 10^5$ km$^2$ (Figure 5i,j) between 2015 and 2020, with the main reduction area concentrated in the southwest (Figure 5e), which belongs to Shigatse, Nagqu and Nagri, administratively. The impact may come from the region's economic development, paddock grazing and fluctuations in natural factors. Grass degradation in the south-western areas is mainly related to overgrazing and climate warming and humidification [46].

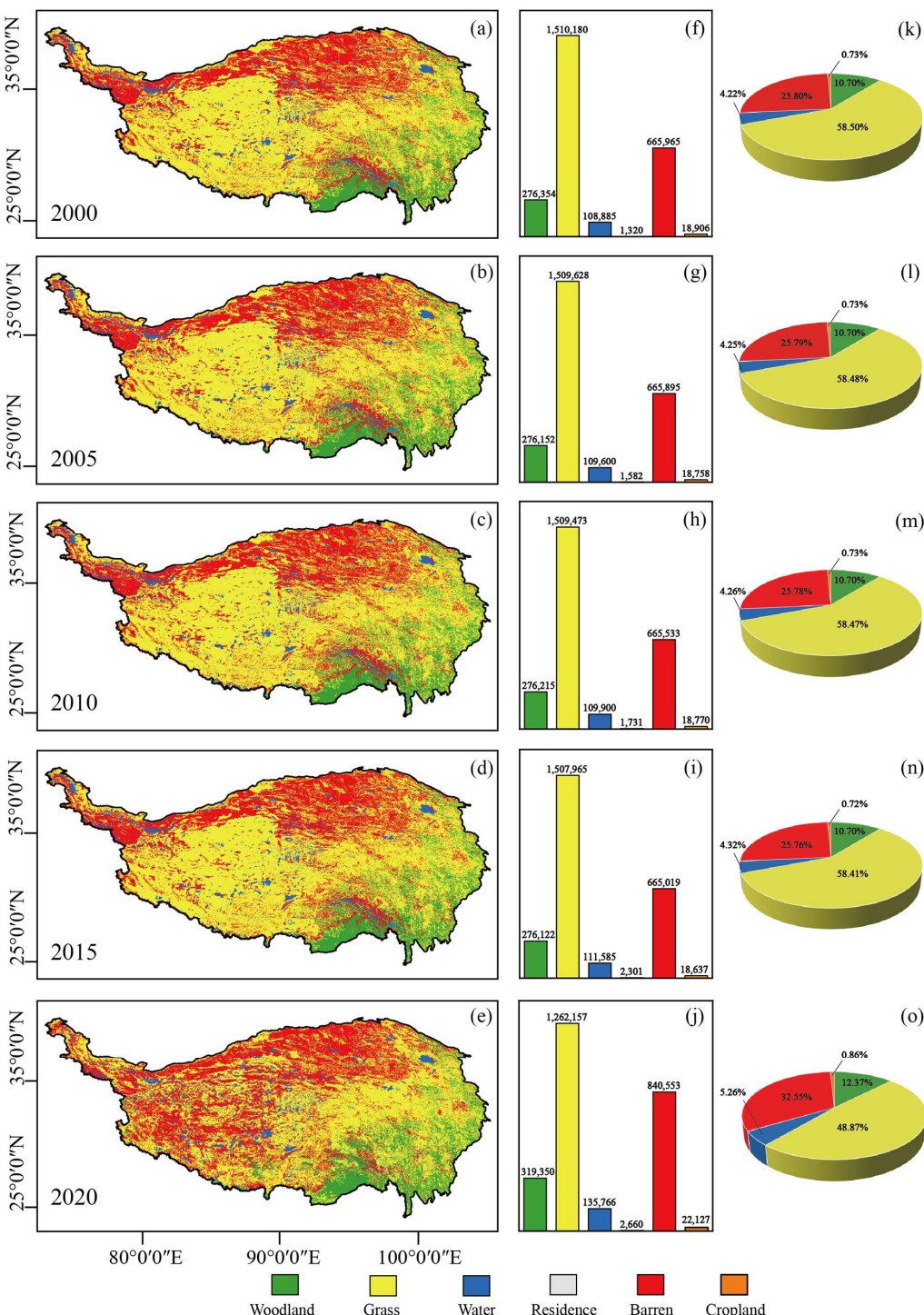

**Figure 5.** Spatiotemporal patterns of LUCC in the Qinghai-Tibet Plateau. The proportion of the areas of residence was less than 0.1% for many years, so it is not shown in the Figure. In order to facilitate statistical analysis and data display, according to its water consumption, the mountain paddy field, hilly paddy field and plain paddy field are uniformly integrated into the paddy. Mountain dryland, hilly dryland, plain dryland and sloping dryland are uniformly integrated into the dryland [47]. (**a–e**) show the temporal and spatial distribution characteristics of LUCC in QTP. Figure 5(f–j) show the total area (km²) of woodland, grass, water, residence, barren and cropland in each studying period. (**k–o**) show the proportion of woodland, grass, water, residence, barren and cropland areas in each studying period.

The spatiotemporal variations of LUCC have important implications for the understanding of distribution characteristics of the RSEI and analyzing its causes [48].

The grass areas are divided into three II land classes according to their coverage, and the proportion of local categories remained stable from 2000 to 2015. The proportion of <20%, 20–50% and >50% was stable at about 34%, 37% and 28%, respectively (Figure 6). However, there was a change in 2020; <20% rose to 47.06%, 20–50% accounted for 37.23%, while >50% decreased to 15.71% (Table 4). It is found that during 2015–2020, the grass areas of QTP experienced degradation. Combined with Figure 5d,e), it can be found that the degradation area is mainly concentrated in the southwest region, whose administrative regions are Nagqu, Shigatse and Naari.

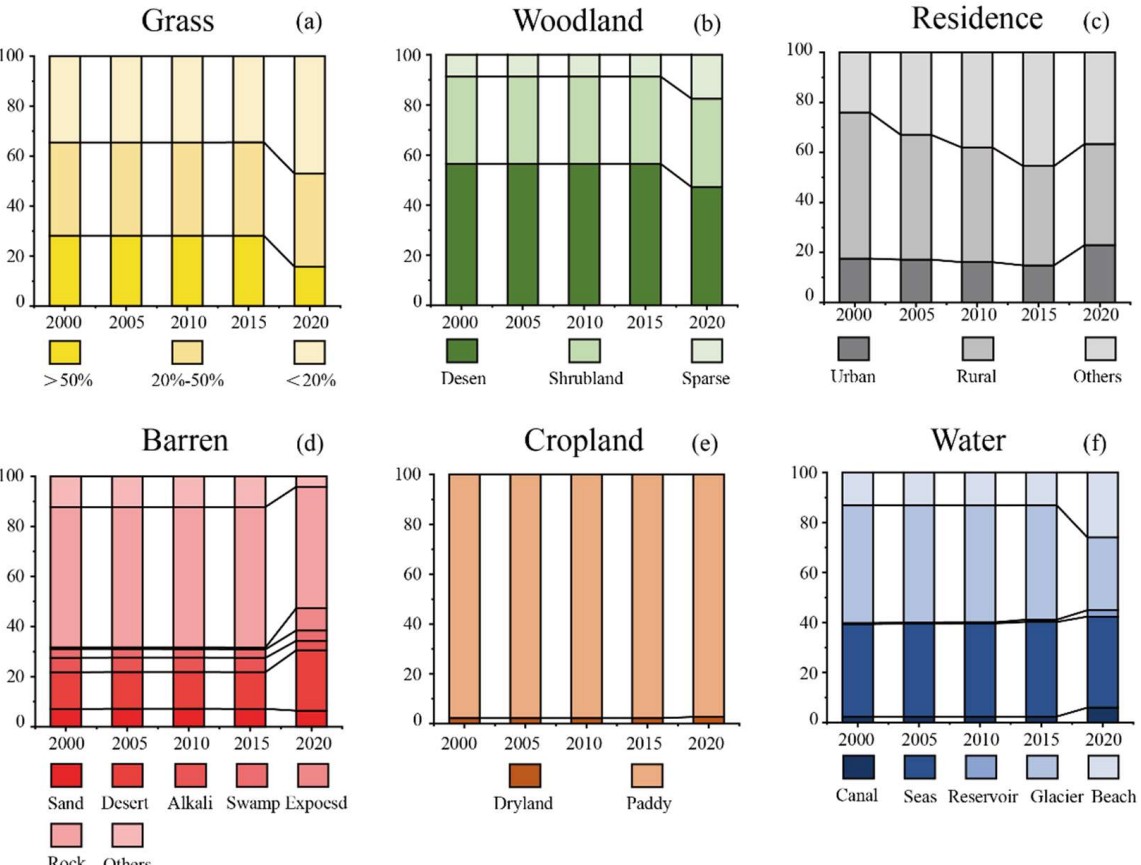

**Figure 6.** Proportion characteristics analysis of LUCC. (**a–f**) mainly show the proportion of class II under class I and the change over time to explore the drivers of RSEI.

The woodland areas are also divided into three II land classes, including dense, shrubland and sparse, according to their coverage, and their proportion characteristics are similar to the grass areas. It shows in (Figure 6) that during 2000–2015, the proportion of dense, shrubland and sparse areas stabilized at about 56%, 35% and 9%, respectively. In 2020, the proportion of dense areas fell to 47.20%, the proportion of shrubland areas remained stable at 35.25%, while the proportion of sparse areas rose to 17.55% (Table 4). Remote sensing images show that the woodland in this area increased from $2.7 \times 10^5$ km$^2$ (Figure 5i) in 2015 to $3.2 \times 10^5$ km$^2$ (Figure 5j) in 2020. However, the transformation of sparse to dense land existed during this period, so the change in the forest cannot be easily judged [33]. Figure 5 shows that the woodland is mainly distributed in the southern low altitude area, because this area is relatively lower and holds abundant water sources.

**Table 4.** Results of proportion analysis of LUCC in the Qinghai-Tibet Plateau.

| LUCC (%) | | Years | | | | |
|---|---|---|---|---|---|---|
| I | II | 2000 | 2005 | 2010 | 2015 | 2020 |
| | Dense | 56.39% | 56.36% | 56.38% | 56.38% | **47.20%** |
| Woodland | Shrubland | 34.85% | 34.87% | 34.88% | 34.88% | 35.25% |
| | Sparse | 8.75% | 8.77% | 8.74% | 8.74% | **17.55%** |
| | >50% | 28.10% | 28.11% | 28.12% | 28.13% | **15.71%** |
| Grass | 20–50% | 37.34% | 37.34% | 37.34% | 37.34% | 37.23% |
| | <20% | 34.56% | 34.55% | 34.54% | 34.52% | **47.06%** |
| | Canal | 2.37% | 2.36% | 2.35% | 2.37% | **5.96%** |
| | Seas | 36.98% | 37.24% | 37.30% | 37.82% | 36.38% |
| Water | Reservoir | 0.42% | 0.44% | 0.46% | 0.95% | **2.56%** |
| | Glacier | 47.12% | 46.80% | 46.65% | 45.76% | **29.15%** |
| | Beach | 13.10% | 13.16% | 13.24% | 13.09% | **25.96%** |
| | Urban | 17.58% | 34.47% | 16.18% | 14.86% | 22.97% |
| Residence | Rural | 58.41% | 49.87% | 45.81% | 39.72% | 40.45% |
| | Others | 24.02% | 32.93% | 38.01% | 45.42% | 36.58% |
| | Sand | 7.12% | 7.22% | 7.22% | 7.19% | 6.42% |
| | Desert | 14.68% | 14.67% | 14.68% | 14.65% | **24.12%** |
| | Alkali | 5.77% | 5.73% | 5.70% | 5.73% | 3.78% |
| Barren | Swamp | 3.46% | 3.41% | 3.40% | 3.37% | 4.15% |
| | Exposed | 0.73% | 0.73% | 0.72% | 0.72% | **8.79%** |
| | Rock | 56.02% | 56.02% | 56.07% | 56.15% | 48.54% |
| | Others | 12.22% | 12.23% | 12.21% | 12.19% | 4.18% |
| Cropland | Dryland | 2.27% | 2.27% | 2.26% | 2.27% | 2.73% |
| | Paddy | 97.73% | 97.73% | 97.74% | 97.73% | 97.27% |

The area of residence is divided into urban, rural and unidentified parts (others) according to the urban-rural distributions, which is gradually increasing (Figure 5f–j). The urban area is gradually rising, reaching 22.97% in 2020 (Table 4), while the rural area is stable and slightly decreasing, from 58.41% in 2000 to 40.45% in 2020. The area of residence has accounted for less than 0.1% of the LUCC for many years (Figure 5k–o), indicating that urbanization in the Qinghai-Tibetan Plateau is proceeding in an orderly way without significant impacts on the ecosystem.

The water area is divided into five class IIs according to the state of matter, including canal, seas, reservoir, glacier and beach. During 2000–2015, canals, seas, reservoirs, glaciers and beaches accounted for about 2.5%, 37%, 0.5%, 46% and 13%, respectively (Table 4). During 2015–2020, glaciers dropped sharply to 29.15%, and while seas remained stable, canals rose to 5.96% and reservoirs rose to 2.56%. The reason for this phenomenon might be the increase in glacial meltwater caused by the rise in temperature (Figure 7) during this period, which greatly replenishes the surface water [28]. In addition, the increase in precipitation is also an important reason for this phenomenon (Figure 8).

The barren areas are divided into six class IIs, including sand, desert, alkali, swamp, exposed, rock and unrecognized areas (others). During 2000–2015, all the classes remained around 7%, 15%, 6%, 4%, 0.7%, 56% and 12.2%, respectively (Table 4). During 2015–2020, desert areas increased to 24.12% and exposed areas increased to 8.79%; the degradation of the grass areas presented in (Figure 5d,e) is strongly associated with this phenomenon.

The cropland has a total of nine class IIs in LUCC. In order to facilitate statistical analysis and data display, according to its water consumption, the mountain paddy field, hilly paddy field and plain paddy field are uniformly integrated into the paddy. Mountain dryland, hilly dryland, plain dryland and sloping dryland are uniformly integrated into the dryland. The dryland and paddy have no great spatiotemporal differences during 2000–2020; the dryland accounted for around 2.5% and the paddy accounted for about

97.7% for many years (Table 4). The cropland has remained below 1% for many years, without a significant impact on the ecological conditions of this area.

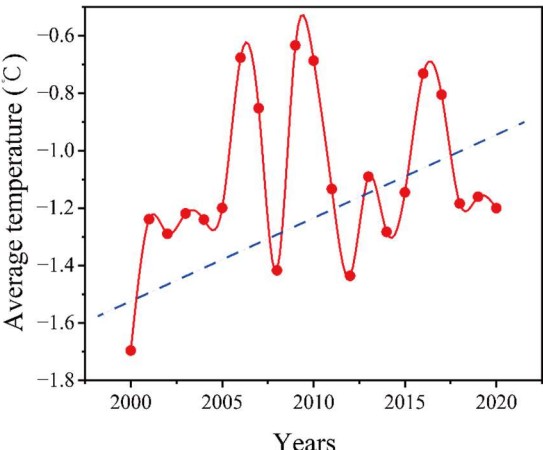

**Figure 7.** The annual temperature in the Qinghai-Tibet Plateau during 2000–2020.

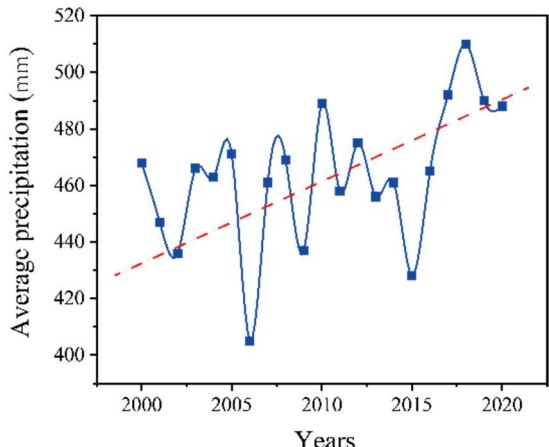

**Figure 8.** The annual precipitation in the Qinghai-Tibet Plateau during 2000–2020.

To sum up, the spatiotemporal characteristics and trend analysis of LUCC shows that the barren areas have been increasing in the QTP during 2000–2020, while the grass areas have been decreasing [49], which may contribute to a degradation in the ecological condition. Although the woodland has also been increasing, there is a shift from dense to sparse areas and it will adversely affect the ecological conditions. The cropland hardly influences ecological condition due to its narrow distribution in QTP.

### 4.1.2. Temperature

Affected by the combined effect of various factors, the temperature and precipitation in the QTP showed a slow increase in fluctuations (Figures 7 and 8). This study takes the average of the whole region based on the multi-year temperature and precipitation raster data. It then combines the monitoring data from the ground weather station as the characterization information of the meteorological factors. The trend of average temperature change in the QTP from 2000 to 2020 is shown in (Figure 7). The minimum is −1.69 °C in 2000 and the maximum is −0.63 °C in 2009, showing an overall upward trend. The average temperature curve holds three peaks, two of which (including the maximum in 2009) occurred during 2005–2010, and the other during 2015–2020, indicating that QTP warmed during both of the periods [50].

The trend analysis (Figure 4b) shows that the ecological condition in the QTP experienced a wide range of degradation during 2005–2010. The analysis results of LUCC

spatiotemporal characteristics (Figure 5b,c) and proportion (Figure 5g,h,l,m) in the same period show that during 2005–2010, the LUCC in the QTP hardly changed, the area and proportion of woodland and grass were not significantly converted to cropland or urban areas, and there is no expansion trend in the impermeable area or the barren areas. Therefore, the decline in RSEI (the declining quality of the ecological conditions) during this period might be caused predominantly by climatic factors [51]. The binding Equation (3) definition of LST shows that the change in LST is dominated by temperature. Taken together, the degradation of the ecological conditions in QTP during 2005–2010 (indicated by RSEI decline) was most likely caused by the increase in temperature.

Furthermore, analysis of LUCC shows that the proportion of glaciers fell to 29.15%, canals rose to 5.96%, and reservoir rose to 2.56% in 2015–2020. This transition is related to the third instance of the maximum temperature during that time period, with continuous warming reaching a specific threshold [52].

### 4.1.3. Precipitation

During the study period, the minimum precipitation was 405 mm in 2006, and the maximum was 510 mm in 2018. There is a long-term increasing trend in precipitation from 2000 to 2020, despite the annual fluctuations (Figure 8).

The temperature curve (Figure 7) shows that the annual temperature in the QTP increased during 2005–2010, and the precipitation curve shows that the average precipitation during this period is lower. There is a certain synergy between the decline in precipitation and the rise in surface temperature, which is reflected in the composition factor of the RSEI; the decline in the WET is accompanied by the rise in the LST. These two indices together contribute to ecological degradation in this period (the decline in the RSEI). The other minimum value of the precipitation appeared in 2016 when the precipitation in this period increased sharply. The proportion characteristics analysis of LUCC (Figure 6) shows that the proportion of WET at a high value reached its maximum during 2015–2020. In addition to the phenomenon of glacial melting supplementing surface water and groundwater due to temperature rises, as found in the LUCC analysis, the sudden increase in precipitation during this period is also an important factor causing the WET to reach its maximum in 2020.

In summary, the gradual rise in temperature and precipitation might be the main cause responsible for the ecological degradation (RSEI index) in the QTP during 2005–2010, and played a positive role in the indices' change during 2015–2020 [53].

### 4.1.4. Grazing

There are two periods (2005–2010 and 2015–2020) of degradation of ecological conditions in the QTP (Figure 3). The previous section mentioned that temperature and precipitation were the main causes of degradation during 2005–2010. The spatiotemporal variation and proportion analysis of LUCC shows that grass degradation is the main factor of ecological degradation during 2015–2020. Figure 5d,e show that the largest change in LUCC in the QTP during 2000–2020 occurred during the sharp decrease in grass areas in the southwest during 2015–2020, while a sharp increase in the barren areas occurred.

The results of the PCA (Table 2) show that NDVI has been the largest contributor to the ecological conditions in the QTP for many years. Therefore, large-scale vegetation degradation has a great impact on ecosystems. Over half of the alpine meadows in the Qinghai-Tibet Plateau (QTP) are degraded due to human activities. Soil degradation from overgrazing is the most direct cause of grassland degradation [54]. The main areas of grass degradation belong administratively to Nagqu, Shigatse and Ngari. The high number of livestock (Figure 9) indicated that the cause of grass degradation in this area might be overgrazing.

To ensure the validity of the casual effect of overgrazing, the meat production of these three regions during 2000–2020 was obtained from the local yearbooks (Figure 10).

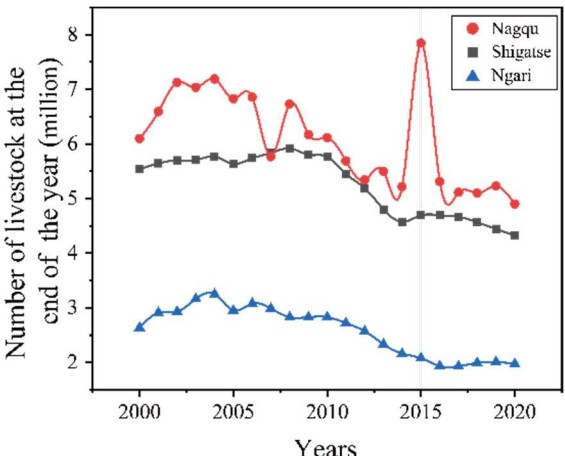

**Figure 9.** The number of livestock in Nagqu, Shigatse and Ngari during 2000–2020.

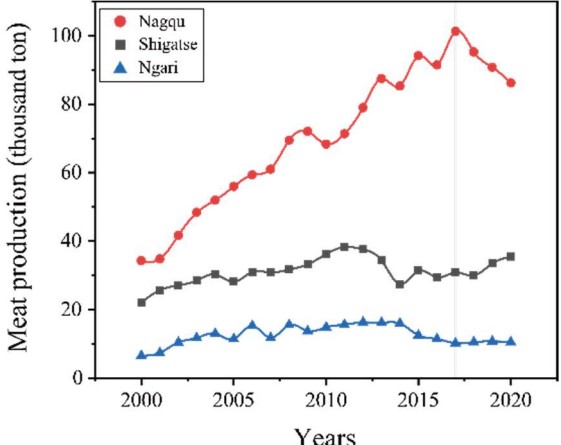

**Figure 10.** The meat production in Nagqu, Shigatse and Ngari during 2000–2020.

Livestock data show that the livestock population in Shigatse is between 4 million and 6 million, gradually decreasing between 2000 and 2020. The number of livestock in Ngari remained between 2 and 3 million, which also showed a gradual downwards trend. The trend in Nagqu was similar to that in the other two regions from 2000 to 2014, with livestock volumes stabilizing between 6 and 7 million. During this period, there was no large-scale grass degradation in the southwest of the QTP. By the end of 2015, the number of livestock in the Nagqu reached 7.84 million, the highest value reported in the past few years (Figure 9).

This sharp increase in the number of livestock could be the main cause of grass degradation in the southwest area of the QTP, which has led to widespread degradation of pastures by exceeding the maximum livestock capacity of the grass areas. Since 2015, the number of livestock in the three regions has been declining, followed by a lack of food caused by the ecological failure to recover, which in turn has led to a decline in livestock [55]. After the livestock population of Nagqu reached its maximum in 2015, the meat production of this area also peaked two years later in 2017 and the maximum value was $1.075 \times 10^7$ t. Then, this trend began to gradually decline, once again confirming that there was a sharp increase in livestock in 2015, which contributes to the degression of grass in this area.

To sum up, the ecological degradation during 2015–2020 is mainly due to the large-scale degradation of grass and the increase in barren land in the southwest area of the QTP. The key impact is that overgrazing in a short period makes it difficult for the ecology to recover, and thus falls into a state of degradation [56].

*4.2. Impact of Human Activities on RSEI*

The spatial distribution characteristics of RSEI show that the area with the lowest ecological conditions in the QTP is mainly the Tsaidam Basin in the north. From the perspective of spatiotemporal characteristics, the area of the land characterized as the worst decreased from $1.6 \times 10^5$ km$^2$ in 2000 to $1.46 \times 10^5$ km$^2$ in 2005 (Figure 3f,g). Since then, it has fluctuated around this value and remained at $1.47 \times 10^5$ km$^2$ until 2020 (Figure 3j). This shows that the ecological conditions have been in a state of dynamic equilibrium after the improvement of 2000–2005. Compared with the spatiotemporal characteristics of LUCC in the same period, it is found that there is no significant change in LUCC during this period (Figure 5f,g). Figure 9 shows that the average temperature of the QTP during this period has remained stable since 2000–2001, during which it rose by 0.4 °C. Figure 10 shows that the average precipitation of the QTP during this period has also been maintained around 460 mm. So, meteorology and LUCC are not the key drivers of this change.

The survey shows that during 1996–2000, the third phase of the Three North Shelterbelt Project was effectively implemented in the Tsaidam Basin. Due to a series of measures, such as afforestation, mountain closure and afforestation, and engineering sand fixation, the sand fixation capacity of the surface has been enhanced, and the regional environment has been significantly improved. These measures are the key reason for the significant ecological recovery of the Tsaidam Basin during 2000–2005. The intervention of artificial measures has made full and effective use of the limited resources of ecologically weak areas, restored vegetation, conserved water sources, and thus increased RSEI.

The conservation process has continued, and during 2001–2010, the fourth phase of the Three North Shelterbelt Project was gradually implemented, and after the exploration and summary of the third phase of the project, the conservation and management measures were more modeled and standardized. In 2016, a new plan was made for the ecological conservation of the Tsaidam Basin according to the wetland conservation and recovery system. However, due to the limited resources of the region itself, it is difficult to obtain further recovery. In the process of economic development, the contradiction between people, LUCC, economic activities and environmental conservation has gradually emerged. Therefore, during 2005–2020, the area of land characterized as the worst did not expand further and remained in a stable state. This area should continue to increase its investment in ecological governance, while introducing advanced technologies and conservation concepts to improve conservation effectiveness.

*4.3. Uncertainty Analysis*

During the acquisition of satellite images, there might have been problems, such as missing data, missing stripes, and inconsistent data resolution. Systematic errors can occur in data processing, noise reduction, and grayscale processing. These problems arise when processing data manually. However, to a certain extent, using GEE to calculate the RSEI to evaluate the ecological condition in QTP could circumvent these issues. Judging from the results, the RSEI is higher in places with rich vegetation and high humidity. In the northern Tsaidam Basin, although the RSEI is low, the ecological condition of the area should be good in the undisturbed state, which is not taken into account by RSEI. In addition, RSEI in this study does not add slope, aspect and water as indices, and the RSEI will be more adaptable if these factors are added. Biodiversity is an important factor in reflecting ecological conditions. In the model upgrade, the impact of anthropogenic activities and biodiversity distribution can also be added, so that RSEI will be more accurate and the evaluation results will hold higher credibility.

**5. Conclusions**

(1)    The ecological condition and environmental quality in the QTP shows a long-term increasing trend, despite the annual fluctuation. Ecological recovery first occurred in 2000–2005, during which the ecological conditions of the whole area increased greatly. Nevertheless, there was a localized decrease in the central area during 2005–2010.

The second period of ecological recovery happened during 2010–2015, with sporadic areas of degradation in the southwest. The second small-scale ecological degradation occurred during 2015–2020.

(2)    The significant ecological recovery and subsequent stability of Tsaidam Basin are the direct effects of the conservation activities. The region should continue to implement conservation measures to prevent ecological degradation. The ecological degradation of the central region during 2005–2010 was caused by the increase in temperature and the decrease in precipitation. Ecological degradation during 2015–2020 was dominated by changes in LUCC, due to grass degradation and decreasing vegetation coverage, due to overgrazing in the southwest. More restricted grazing policies should be implemented in this area to ensure a sustainable grazing industry. The comparison of the magnitude of the two periods of ecological degradation suggested that the influence of natural factors is greater. Human factors are related to the distribution characteristics of the region's vast population.

**Author Contributions:** S.W. and G.L. contributed equally to guide this work. J.C. and E.W. are coauthors and contributed equally to this work. Conceptualization, J.C. and E.W.; methodology, J.C., E.W., R.F., L.X., S.W. and G.L.; software, J.C., E.W. and R.F.; validation, S.W. and G.L.; formal analysis, J.C., E.W., R.F., K.N., L.X. and L.Y.; investigation, J.C., E.W., Y.L., K.N., R.L., X.X., J.Z. and R.F.; resources, E.W.; data curation, J.C. and E.W.; writing—original draft preparation, J.C., Y.L. and R.F.; writing—review and editing, S.W., Y.L. and G.L.; visualization, J.C., J.Y. and R.F.; supervision, S.W. and G.L.; project administration, S.W., and G.L.; funding acquisition, S.W. All authors have read and agreed to the published version of the manuscript.

**Funding:** This research was funded by the Regulation for assessment of eco-environmental damage in protected areas (No. 2021-LY-025).

**Data Availability Statement:** Not applicable.

**Conflicts of Interest:** The authors declare no conflict of interest.

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
