# Peer review of "Spatiotemporal Dynamics of Ecological Condition in Qinghai-Tibet Plateau Based on Remotely Sensed Ecological Index"

_remotesensing, doi:10.3390/rs14174234_

Round 1

Reviewer 1 Report

Dear Authors, I have read your research manuscript very carefully and I believe that there are some issues that need to be addressed before being considered for publication.

It is not clear whether RSEI (remote sensing ecological index) is a new index proposed by the authors or an already known index. In the later case the authors should provide references. If this is a new index I think that authors should give it a name as many different indices are found in the literature and all are called RSEI!!!!!! Maybe you should refer to other similar RSEI in the introduction.

I believe that the authors should rewrite the methods section as the methodology is not clear. I feel that you calculated the proposed index 5 times, every five years. Is that the case?

How many NDVIs did you use for every period (Jul to Sep)? One every 16 days? did you calculate an average NDVI for every month? Or an average NDVI per year? (The same questions are for all the indices you used).

What was the cloud coverage in these images? Usually, in such areas with high altitude the cloud-covered pixels in MODIS products are masked.

There is twice the section title “3.3. The four contributing indices”

Finally, I wonder: The results would be different if you used only NDVI (as it is connected to meteorological conditions) or NDVI coupled with LST?

would it make a difference if you used a geo-morphological indicator (such as slope or slope length)?

Is your RSEI better than the others proposed in the literature? In what sense?

Best regards

Reviewer 2 Report

Dear Authors

Please have a look at my suggestions to improve the quality of the manuscript.

Line-19: QTP insists QTB

Line 33: what was the fact the degradation of overgrazing in the southwest? Is it any factors suggesting that degradation is getting higher due to increase livestock flocks, or another factors? 

Line 38: please re-check or consult with experts related to your (3rd) findings 

Line 143: add the link of CAS platform

Please add the codes of GEE for additional materials 

Reviewer 3 Report

Review

The article is devoted to the study of the spatial and temporal dynamics of the ecological condition of the Qinghai-Tibet Plateau based on the Remote Sensing Ecological Index (RSEI), which is a combination of four indices: NDVI, WET, LST and NDBSI.

As the reasons for the ecological degradation in the QTP, the authors consider both natural factors - an rise temperature and a change the precipitation, as well as anthropogenic, economic and political factors.

The authors describe in sufficient detail and step by step all the indices (methods) used in the scientific work. The use of the above indices in combination when assessing the ecological condition of the territory under consideration for the specified period (2000-2020) increases the quality and representativeness of the work.

The researchers identified several stages of improvement and / or degradation of the ecological condition at QTP, indicating possible causes and recommendations for environmental measures, which is the strength of the work. The results of the work can be useful for both GIS specialists and environmental organizations.

The article is well structured, the authors have done a deep analysis of the spatiotemporal change of each index, the conclusions are logical, in general, the study is presented quite well and can be recommended for publication.

Round 2

Reviewer 2 Report

Dear Authors

The manuscript has been modified after suggestions and as a reviewer allow me to recommend you to submit another journal related MDPI platforms (e.g.Land)

Thanks

Author Response

Your comments are very informative, thanks.
